# Apical targeting of the formin Diaphanous in *Drosophila* tubular epithelia

Tal Rousso[1], Annette M Shewan[2†], Keith E Mostov[2], Eyal D Schejter[1*], Ben-Zion Shilo[1*]

[1]Department of Molecular Genetics, Weizmann Institute of Science, Rehovot, Israel; [2]Department of Anatomy, University of California, San Francisco, San Francisco, United States

**Abstract** Apical secretion from epithelial tubes of the *Drosophila* embryo is mediated by apical F-actin cables generated by the formin-family protein Diaphanous (Dia). Apical localization and activity of Dia are at the core of restricting F-actin formation to the correct membrane domain. Here we identify the mechanisms that target Dia to the apical surface. PI(4,5)P2 levels at the apical membrane regulate Dia localization in both the MDCK cyst model and in *Drosophila* tubular epithelia. An N-terminal basic domain of Dia is crucial for apical localization, implying direct binding to PI(4,5)P2. Dia apical targeting also depends on binding to Rho1, which is critical for activation-induced conformational change, as well as physically anchoring Dia to the apical membrane. We demonstrate that binding to Rho1 facilitates interaction with PI(4,5)P2 at the plane of the membrane. Together these cues ensure efficient and distinct restriction of Dia to the apical membrane.

*For correspondence: eyal.
schejter@weizmann.ac.il (EDS);
benny.shilo@weizmann.ac.il
(B-ZS)

†Present address: School of
Chemistry and Molecular
Biosciences, University of
Queensland, Brisbane, Australia

Competing interests: The
authors declare that no
competing interests exist.

Reviewing editor: Helen
McNeill, The Samuel Lunenfeld
Research Institute, Canada

## Introduction

Epithelial cells that comprise tubular organs are highly polarized, a feature that enables them to execute functions such as vectorial secretion and absorption of nutrients. Polarization is apparent in the distinct composition of membrane domains: the apical membrane—the surface facing the lumen, the basal membrane, which contacts the underlying extracellular matrix (ECM), and the lateral surfaces, which contain specialized cellular junctions that adhere adjacent cells (*Bryant and Mostov, 2008*). The generation and maintenance of cell polarity is achieved by domain-specific proteins and lipids, which support the unique organization and function of each region. Among these, asymmetric distribution of phosphoinositides has been shown to be crucial for membrane identity and lumen formation in tubular systems. PI(4,5)P2 in the apical surface and PI(3,4,5)P3 in the basal membrane have been shown to tether specific polarity and cytoskeleton related proteins, which define their respective domains (*Martin-Belmonte and Mostov, 2007*).

Cytoskeletal structures play key guidance roles underlying maintenance of epithelial cell polarity. They perform these functions by serving as membrane scaffolds, supporting adhesion, and enabling vesicle transport (*Nance and Zallen, 2011*; *Tepass, 2012*). One such structure, which is a common feature of tubular tissues, is a network of actin microfilaments lining the apical surface of the tube cells. In a previous study we found that in *Drosophila* tubular organs, this network mediates myosinV based transport of vesicles, promoting their secretion from the apical surface into the tube lumen. The actin-nucleator responsible for generating these structures was shown to be the formin-family protein Diaphanous (Dia) (*Massarwa et al., 2009*).

Apical restriction of Dia activity in this context is the consequence of tight apical localization of the Dia protein, which was shown to be a common feature of all epithelial cells generating the different *Drosophila* embryonic tubular organs. Thus, apical targeting of Dia is at the core of a cellular mechanism

**eLife digest** Many physiological processes are directional, which means that tissues and organs often need a sense of spatial orientation in order to function properly. In most tissues, this sense of direction relies on certain proteins and infrastructure components of the cell being located in specific subcellular regions, rather than being distributed in a more symmetrical fashion throughout the cell: the latter phenomenon is known as cell polarity.

Exocrine tissues (that is, glands) are composed of tubular epithelial cells organized around a central lumen: the cells in the gland secrete various products (such as enzymes) into the lumen, so that they can be carried to the target organ elsewhere in the body. Epithelial cells in these tissues are therefore polarized to enable directional transport to the lumen. An example of cell polarity is a network of actin filaments that lines the apical surface of these cells (the surface nearest the common lumen). This actin network helps to shuttle cargo to the lumen by assisting with directional, coordinated secretion, among other processes.

In fruitflies, the construction of the apical actin network depends on the presence of a protein called Diaphanous. However, the signals that lead to the localization of this protein near the apical membrane of the cells are not well understood. Now Rousso et al. report that a modified lipid, called PI(4,5)P$_2$, is involved in this localization. However, they also show that this lipid does not govern the apical localization of Diaphanous on its own: rather, an enzyme called Rho1 must also be present to assist with the localization of Diaphanous and to ensure that actin is deposited in the correct place. Rousso et al. also demonstrate that PI(4,5)P$_2$-mediated localization of *Drosophila* Diaphanous occurs in mammalian cells. Lipid-protein collaboration also targets other proteins to the apical membrane. A common mechanism may therefore underlie cell polarity in tubular organ tissues in flies and mammals.

generating actin cables that emanate from the apical membrane, and enable apical secretion. While the delivery of apical and baso-lateral transmembrane proteins through specialized routes of the secretory pathway has been well studied (*Weisz and Rodriguez-Boulan, 2009*), much less is known about the targeting of cytoplasmic proteins such as Dia to distinct membrane domains.

Dia belongs to the formin family of actin nucleators, which regulate the formation of linear actin cables. The Dia-related formins (DRFs) can be functionally divided into two major domains, each encompassing roughly one half of the protein sequence (*Goode and Eck, 2007*). The C-terminal portion of DRFs regulates actin polymer assembly by mediating microfilament nucleation, elongation and processive capping. Key functional sub-domains include the FH2 domain, which acts as a dimer, and moves processively with the growing barbed end, and the FH1 domain, which together with profilin acts to accelerate filament elongation by recruiting monomeric actin. The N-terminal portion of DRF nucleators is regulatory, governing the activation state of the molecule through interactions with various effectors. Importantly, this region has been shown to play significant roles in directing DRF localization in vivo (reviewed in *Higgs, 2005*; *Chesarone et al., 2010*). DRFs are autoinhibited due to an intra-molecular interaction between the C-terminal DAD domain and the N-terminal DID domain, which maintains the molecule in a closed conformation. Upon binding of GTP-bound Rho1 to the N-terminal GTPase-binding domain, autoinhibition is relieved, allowing Dia to assume an open, active conformation and promote actin polymerization (*Lammers et al., 2005*; *Otomo et al., 2005*; *Rose et al., 2005*).

DRFs have been shown to play critical roles in the formation of diverse actin-based structures in many cellular contexts, including cytokinesis, cell adhesion, polarized cell growth and cell migration (*Faix and Grosse, 2006*). Accordingly, Dia-like proteins assume distinct subcellular localizations in a cell type specific manner. To prevent spatially unrestricted actin assembly in cells, localization of DRFs is highly regulated, and involves different factors and mechanisms. For example, the yeast formin Bni1p localizes during budding to the mother cell bud neck, by binding to the polarity factor Spa2, via a mechanism that involves four different localizing domains (*Liu et al., 2012*). Targeting of mDia2, a mammalian homologue of Dia, to the cleavage furrow during cytokinesis, is achieved by binding to the scaffold protein anillin (*Watanabe et al., 2010*). In *Drosophila*, Dia is recruited to lamellipodia by the actin regulator Ena/Vasp in a small subset of epidermal cells, during the process of embryonic

dorsal closure (**Homem and Peifer, 2009**). In *Drosophila* tubular structures, *dia* mRNA is tightly concentrated at the apical side of the cells, by a mechanism that was shown to be distinct from the Dia protein apical targeting mechanism (**Massarwa et al., 2009**). Thus, targeting of Dia-like proteins is regulated at multiple levels, to assure restriction of activity to the correct cellular domain.

Here we examine the mechanisms that mediate apical targeting of the Dia protein in tubular organs of *Drosophila*. We find that efficient apical targeting of Dia requires the detection of both PI(4,5)P2 and Rho1 on the apical surface. We show, both in a mammalian cell culture model and by genetic manipulations in *Drosophila* tubular organs, that PI(4,5)P2 levels regulate Dia localization. An N-terminal basic domain of Dia is critical for this regulation, indicative of an electrostatic-based direct interaction between Dia and PI(4,5)P2. We demonstrate that apical enrichment of PI(4,5)P2 is a common feature of tubular organs in the fly, which can be attributed in part to the apical restriction of the PIP5 kinase (PIP5K) Skittles. We further show that Rho1 facilitates Dia apical targeting, both by inducing the Dia open conformation, thereby exposing N-terminal domains that are critical for localization, as well as by physically anchoring Dia to the apical surface. This anchoring appears to promote the Dia–PI(4,5)P2 interaction, leading to synergistic effects of these two apical cues. We therefore propose that a multi-tiered targeting mechanism, utilizing both global and specific features of tubular organs, achieves a distinct localization pattern of Dia, thereby enabling execution of the cardinal process of apical secretion in these organs.

## Results

### The open conformation of Dia facilitates apical targeting through N-terminal domains

Endogenous Dia exhibits a highly polarized distribution, accumulating at the apical surface of epithelial cells in diverse tubular tissues of the *Drosophila* embryo (**Figure 1A–C**; **Massarwa et al., 2009**). To dissect and quantitatively assess the structural basis for this polarized distribution, we monitored the localization patterns of different GFP-tagged Dia constructs (**Figure 2A**), following tissue-specific expression in several tubular organs. These constructs do not contain the 3′UTR sequences mediating RNA localization, and thus reflect targeting at the protein level. Upon expression at similar levels to endogenous Dia (**Figure 1—figure supplement 1**), a full-length Dia construct (GFP-Dia-FL) displayed an apically polarized distribution, albeit less dramatic than the endogenous protein (**Figure 1D–F**). Measurements of GFP intensity in embryonic salivary gland cells further substantiated the enrichment of GFP-Dia-FL at the apical surface, relative to both the cell cytoplasm and to the septate junction (SJ) domain, representing the lateral membrane (**Figure 1J–L** and **Figure 1—figure supplement 1**).

Removal of the extreme C-terminus of Dia, which includes the DAD domain (GFP-Dia-ΔDAD), generates an open form of Dia, which is relieved from auto-inhibition (**Li and Higgs, 2003**). While expression of this construct in tubular epithelia results in protein levels similar to the full-length form (**Figure 1—figure supplement 1**), GFP-Dia-ΔDAD displays a much tighter apical localization pattern (**Figure 1G–I**), suggesting that the molecule is now more accessible to apical-targeting signals. Quantification demonstrates that apical enrichment of GFP-Dia-ΔDAD is enhanced up to fourfold relative to GFP-Dia-FL (**Figure 1J–L**). Further measurements in GFP-Dia-ΔDAD expressing cells reveal that ~45% of the total GFP intensity in the cell is concentrated at the apical surface, which encompasses only ~5% of the total cell area. When compared with the GFP-Dia-FL expressing cells, a clear shift in the GFP distribution is observed from the cytoplasm to the apical domain. In contrast, a similar degree of localization to the lateral membrane SJ domain is exhibited by both constructs (**Figure 1—figure supplement 1**). Thus, enhanced apical localization of GFP-Dia-ΔDAD appears to result primarily from efficient apical recruitment of its cytoplasmic pool. The intensity differences between the apical and cytoplasmic domains were therefore used in this study as the primary measure for apical enrichment of Dia constructs.

To identify the domains within Dia that mediate apical localization of the open form, we examined the localization patterns of different reporter constructs (**Figure 2A**). While these assays focused on the embryonic salivary gland, qualitatively similar observations were made in other tubular organs (**Figure 2—figure supplement 1**). We initially examined separately the localizing activity of the two halves of the Dia-ΔDAD protein, each of which harbors distinct functional domains. A construct containing only the C-terminal half (GFP-Dia-C′) was distributed within the cytoplasm upon expression in tubular epithelia (**Figure 2B** and **Figure 2—figure supplement 1**). In contrast, the reciprocal

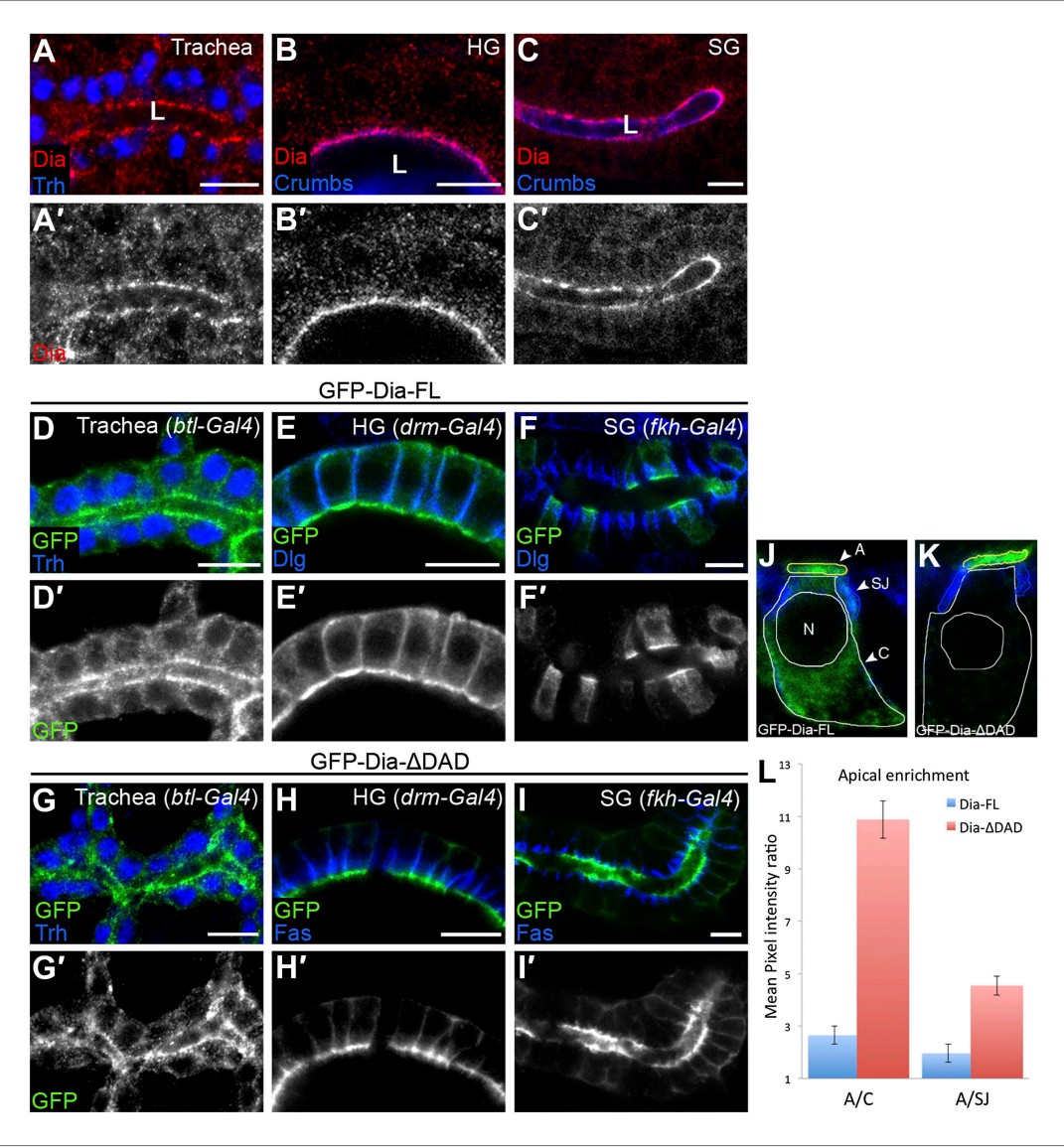

**Figure 1**. The open conformation enhances Dia apical localization. (**A–C'**) Endogenous Dia staining (red, shown separately [grey] in single-primed panels) shows apical localization in the trachea (**A**), hindgut (HG, **B**) and salivary gland (SG, **C**) of stage 15 embryos. Trh (blue, **A**) marks nuclei of tracheal cells. Crumbs (blue, **B** and **C**) marks the apical surface. Apical surfaces face the lumen (L). (**D–I'**) The localization of GFP-tagged Dia constructs (***Figure 2A***, green and shown separately [grey] in single-primed panels) was monitored following expression in stage 14–15 embryonic trachea, hindgut and salivary gland under *btl-Gal4* (**D** and **G**), *drm-Gal4* (**E** and **H**) and *fkh-Gal4* (**F** and **I**), respectively. Trh (blue, **D** and **G**) marks nuclei of tracheal cells. Dlg and FasIII (blue, **E,F,H,I**) mark septate junctions. GFP-Dia-ΔDAD (**G–I'**) is more apically localized than GFP-Dia-FL (**D–F'**). Scale bars, 10 μm. (**J–L**) Quantification of GFP fluorescence-intensity ratios. All measurements were carried out on confocal images of the different constructs expressed in the salivary gland. (**J** and **K**) Illustration of the different cellular domains measured in GFP-Dia-FL (**J**) and GFP-Dia-ΔDAD (**K**) expressing salivary gland cells. A–apical domain (outlined in yellow), SJ–septate junctions (outlined in blue), C–cytoplasm (outlined in white), N–nucleus. (**L**) Apical enrichment quantification represented by the mean pixel intensity ratio between the apical domain and the cytoplasm (A/C) or the septate junctions region (A/SJ). Error bars represent standard error. n = 20. See also ***Figure 1—figure supplement 1***.

The following figure supplements are available for figure 1:

**Figure supplement 1**. Characteristics of Dia apical enrichment.

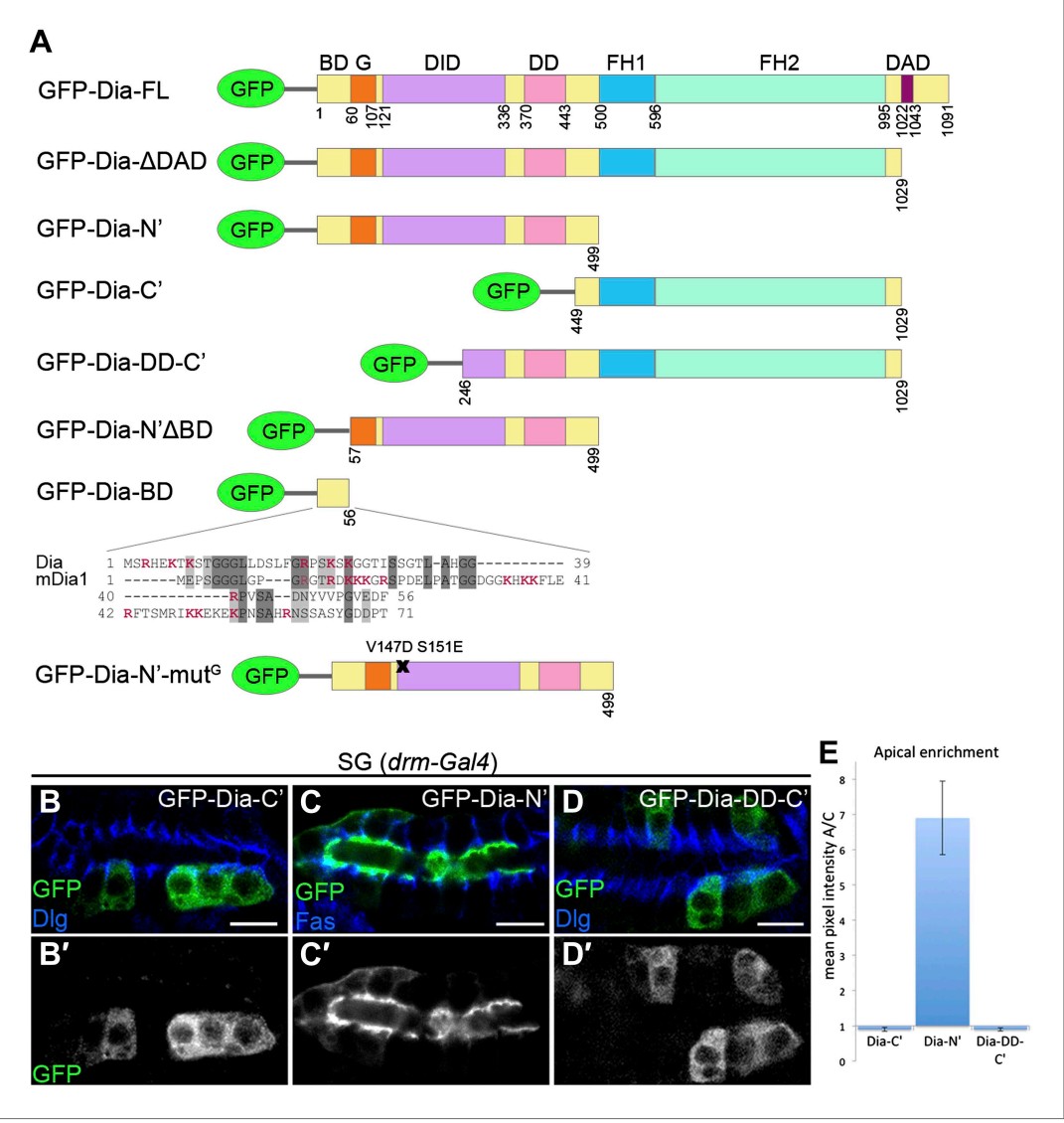

**Figure 2**. The N-terminal domain of Dia is necessary and sufficient for apical targeting. (**A**) Schemes of Dia constructs tagged with GFP at the N-terminus (**Homem and Peifer, 2009** and this study, see 'Materials and methods'). Shown for GFP-Dia-BD is a global pairwise amino acid sequence alignment of the basic domain (BD) between *Drosophila* Dia and the mammalian homologue mDia1 (Needle, EMBOSS package, default parameters). Basic residues are marked in red. BD, Basic Domain; G, GTPase-binding domain; DID, Diaphanous Inhibitory Domain; DD, Dimerization Domain; FH1 and FH2, Formin Homology domains 1 and 2; DAD, Diaphanous Autoregulatory Domain. (**B–D'**) Stage 15 embryonic salivary glands expressing GFP-tagged Dia constructs (**A**) under *drm-Gal4* (green, shown separately (grey) in single-primed panels). Dlg and FasIII (blue) mark septate junctions. (**B** and **B'**) The GFP-Dia-C' construct displays cytoplasmic distribution. (**C** and **C'**) The GFP-Dia-N' construct is highly restricted to the apical domain. (**D** and **D'**) The dimerization domain (DD) does not contribute to apical targeting of the Dia-C' domain, as GFP-Dia-DD-C' distribution is completely cytoplasmic (compare with **B**). Some cells appear to contain two nuclei, which may reflect an effect of this constitutively-activated Dia construct on cytokinesis. Scale bars, 10 μm. (**E**) Quantification of GFP fluorescence intensity in (**B–D**), represented by the mean pixel intensity ratio between the apical domain and the cytoplasm. Error bars represent standard error. n = 7–11. See also **Figure 2—figure supplement 1**.

The following figure supplements are available for figure 2:

**Figure supplement 1**. Dia constructs show similar localization patterns in all tubular organs.

N-terminal construct (GFP-Dia-N') displayed striking apical enrichment, reminiscent of the localization pattern of the intact GFP-Dia-ΔDAD open molecule (*Figure 2C* and *Figure 2—figure supplement 1*). Indeed, quantification of relative fluorescence levels demonstrated strong apical enrichment of GFP-Dia-N', while no apical bias whatsoever was observed for GFP-Dia-C' (*Figure 2E*). These observations suggest that the N-terminal half of Dia is both necessary and sufficient for apical targeting.

Since the Dia protein functions as a dimer, one concern is that the localization of the tested constructs will be influenced by dimerization to the endogenous, full-length protein. However, when the dimerization domain (DD) was added to the Dia-C' construct (GFP-Dia-DD-C'), no apical localization was detected (*Figure 2D–E* and *Figure 2—figure supplement 1*), demonstrating that apical localization of Dia constructs is not biased by dimerization to endogenous Dia.

Taken together, this structure–function analysis supports a model whereby the open conformation of Dia facilitates efficient apical localization by exposing critical domains to targeting cues. The Dia-targeting regions are restricted to the N-terminal half of the protein.

## PI(4,5)P2 directs apical targeting of *Drosophila* Dia in MDCK cysts

The striking difference in the apical distribution of Dia between tubular organs and other non-polar tissues suggested the presence of an inherent apical cue, unique to tubular epithelial tissues. Phosphoinositides are general regulators of membrane identity in tubular tissues. In 3D cysts of cultured Madin–Darby canine kidney (MDCK) cells, a well-established system for recapitulation of epithelial tube morphogenesis, PI(4,5)P2 is a key determinant of the apical membrane, where it is highly enriched (*Martin-Belmonte et al., 2007*). We therefore utilized the MDCK cyst system, in which phosphoinositide levels can be readily manipulated, to examine a possible role for PI(4,5)P2 in the regulation of Dia localization.

Both the full-length and open forms of *Drosophila* Dia, tagged with GFP, were stably expressed in MDCK cyst culture, which were subsequently FACS-sorted, to ensure selection of cysts displaying similar GFP expression levels. The fluorescent GFP signal allowed us to monitor localization of these constructs following cyst maturation. *Drosophila* GFP-Dia-FL failed to localize to a specific subcellular domain, and was homogenously distributed in the cytoplasm of cyst cells (*Figure 3A*). In marked contrast, *Drosophila* GFP-Dia-ΔDAD was targeted apically, co-localizing with the apical marker gp-135 (*Figure 3B*).

The localization features of the open form of *Drosophila* Dia were therefore recapitulated in the heterologous MDCK system, consistent with the notion that apical targeting is mediated by a conserved element such as PI(4,5)P2. To directly test the involvement of PI(4,5)P2 in Dia localization, we delivered exogenous PI(4,5)P2 to the basal surface of cysts expressing GFP-Dia-ΔDAD (*Gassama-Diagne et al., 2006*), and monitored the effects on its localization pattern. GFP-Dia-ΔDAD localization was dramatically altered, even after very short treatments of 10 min, displaying redistribution from the apical to the basal surface, while remaining apical in mock treated cysts (*Figure 3C–F*). The rapid shift in GFP-Dia-ΔDAD localization preceded the general shift in polarity induced by exogenous PI(4,5)P2 treatment in this system (*Martin-Belmonte et al., 2007*). This can be readily appreciated by the partial basolateral localization of the apical membrane marker gp-135, which no longer co-localized with GFP-Dia-ΔDAD following treatment (*Figure 3E,F*). The rapid nature of Dia re-localization suggests that recruitment to the basolateral membranes is a direct result of exogenous PI(4,5)P2 treatment. These observations imply that PI(4,5)P2 could fulfill a prominent role in the regulation of Dia apical targeting in tubular structures.

## PI(4,5)P2 and Skittles, a PIP5-kinase, are apically enriched in *Drosophila* tubular organs

We next asked whether PI(4,5)P2 also serves as an apical cue for Dia localization in *Drosophila* tubular organs. To follow the endogenous distribution of PI(4,5)P2, we expressed PH-PLCδ-GFP, a PI(4,5)P2 sensor in which the PH domain of PLCδ is tagged with GFP (*Downes et al., 2005*), in different tubular organs. Imaging the GFP fluorescence in live tissues showed significant apical enrichment of the sensor in the embryonic hindgut and larval salivary gland cells (*Figure 4B,D*), which was apparent but less pronounced in the embryonic trachea (*Figure 4C*). Apical membranes of *Drosophila* tubular epithelia are therefore enriched in PI(4,5)P2, as in mammalian systems. In general, imaging of these fine subcellular domains in live samples produced clearer and more consistent localization patterns than fixed preparations, leading us to use the live imaging approach whenever applicable, throughout the analysis.

Local production of PI(4,5)P2 at the apical membrane by PIP5-kinases could serve as a means for enrichment at this site. Skittles (Sktl), one of the two *Drosophila* PIP5-kinases which catalyze PI(4,5)P2

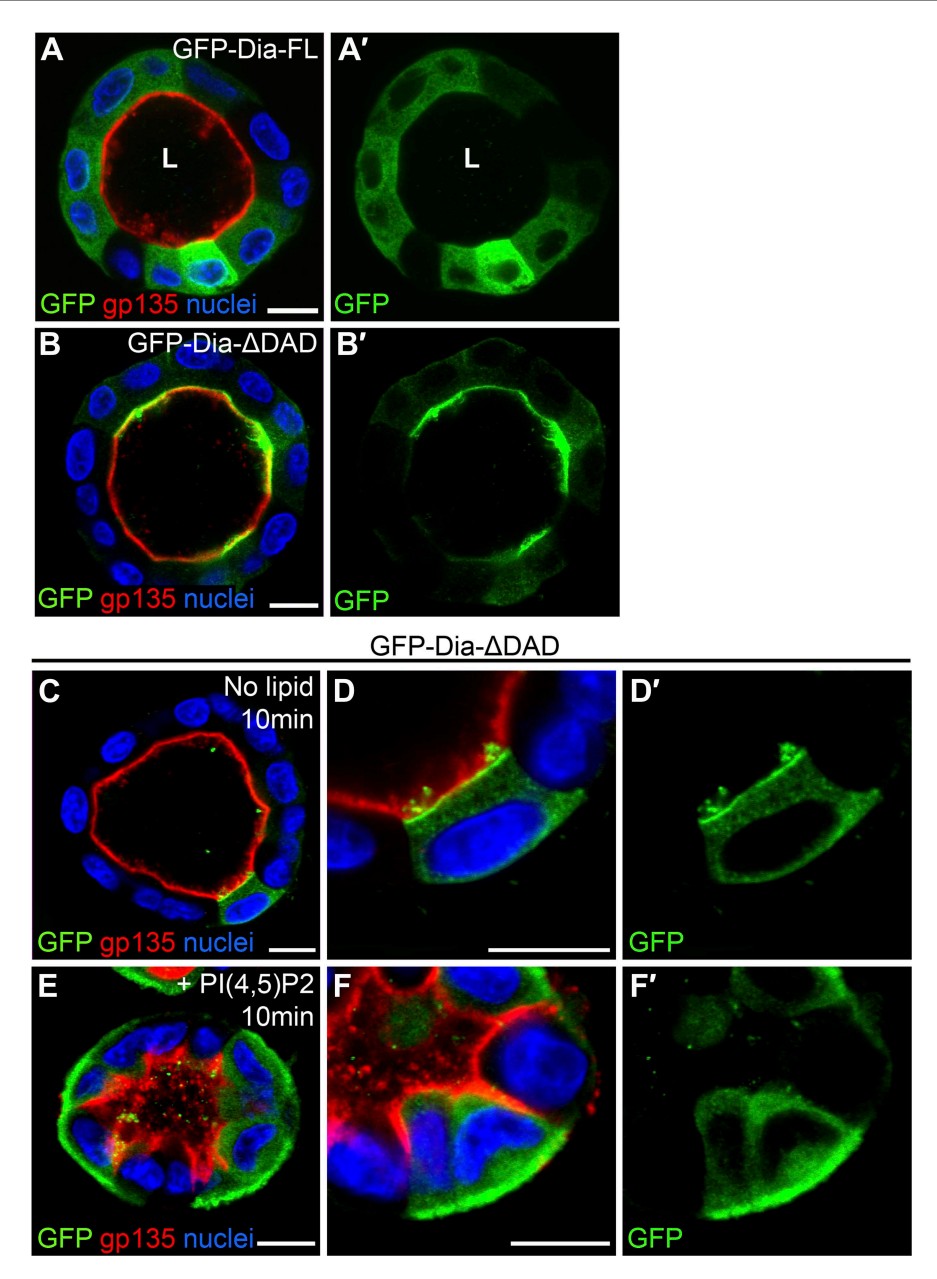

**Figure 3**. PI(4,5)P2 levels regulate *Drosophila* Dia localization in MDCK cysts. (**A–B'**) The localization of GFP-tagged Dia constructs (green, shown separately in single-primed panels) was monitored following stable expression in MDCK cyst culture. GFP-Dia-FL (**A**) displays an entirely cytoplasmic distribution, while GFP-Dia-ΔDAD (**B**) is highly restricted apically. (**C–F'**) GFP-Dia-ΔDAD expressing MDCK cysts were incubated with histone (no lipid, **C–D'**) or PI(4,5)P2-histone complexes (+PI(4,5)P2, **E–F'**) for up to 10 min. Exogenous PI(4,5)P2 re-localized GFP-Dia-ΔDAD to the basal membrane (**E** and magnified in **F**) in comparison to control cysts (**C** and magnified in **D**). Apical enrichment is slightly reduced in the control cysts (compare **C** with **B**), probably due to pre-incubation with Trypsin and low-calcium levels. Nuclei are stained with Hoechst. Apical surfaces face the lumen (L). Scale bars, 10 μm.

production from PI(4)P (*Figure 4A*), was shown to be critical for PI(4,5)P2 production in various systems (*Gervais et al., 2008*; *Fabian et al., 2010*), and is endogenously expressed in different *Drosophila* embryonic tubular organs (BDGP-Berkeley *Drosophila* Genome Project). Indeed, when we expressed an RFP-tagged version of Sktl in *Drosophila* tubular organs, it was targeted predominantly to the apical

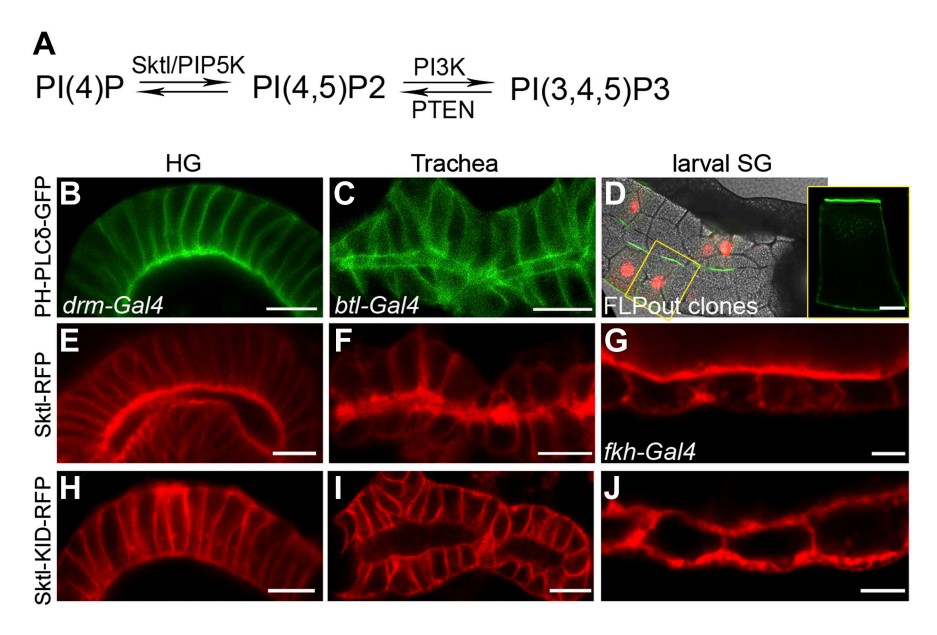

**Figure 4**. PI(4,5)P2 and Skittles, a PIP5 kinase, are apically enriched in *Drosophila* tubular organs. (**A**) Simplified scheme of the PI(4,5)P2 biosynthetic pathway. PI(4)P is phosphorylated by Sktl /PIP5K to generate PI(4,5)P2. PI(4,5)P2 can be further phosphorylated by PI3K to produce PI(3,4,5)P3, which can be dephosphorylated by PTEN phosphatase to regenerate PI(4,5)P2. (**B–D**) Live imaging of PH-PLCδ-GFP following expression in stage 14 embryonic hindgut (**B**) and trachea (**C**) under *drm-Gal4* and *btl-Gal4*, respectively, and in third instar larval salivary gland FLP-out clones (**D**). (**D**) Salivary gland FLP-out clones expressing PH-PLCδ-GFP under *actin-Gal4*. Clones are marked by nuclear RFP (red), and the salivary gland outline is visualized using transmitted light. Inset shows sensor distribution within a single magnified clone cell. PH-PLCδ-GFP is enriched at the apical surface of these tubular organs. (**E–J**) Live imaging of Sktl-RFP and Sktl-KID-RFP, following expression in the embryonic hindgut (**E** and **H**) and trachea (**F** and **I**) under *drm-Gal4* and *btl-Gal4*, respectively, and in the 2nd instar larval salivary gland under *fkh-Gal4* (**G** and **J**). Sktl is enriched at the apical domain, while a kinase-dead form (Sktl-KID) is localized throughout the cell cortex. Scale bars, 10 μm, and 20 μm (**D**).

surface of cells (*Figure 4E–G*). Thus, apical targeting of Sktl may account for PI(4,5)P2 accumulation at this membrane.

It was proposed that PIP5-kinases are targeted to the membrane through electrostatic interactions between basic residues in their sequence and acidic lipids on the membrane, including PI(4,5)P2 itself (*Kwiatkowska, 2010*). Supporting this notion is the observation that a mutated form of PIP5-kinase, containing an inactive kinase domain, fails to localize to the same membrane domain as the wild-type protein (*Fairn et al., 2009*), demonstrating that continuous PI(4,5)P2 generation is required to enhance the localization bias of PIP5-kinases. We find a similar relationship in *Drosophila* tubular organs between Sktl activity and apical localization: a mutated form of Sktl, where the kinase domain is inactive (Sktl-KID), was no longer enriched apically, and instead assumed a general cortical distribution (*Figure 4H–J*). We assume that the excess of Sktl-KID molecules may sequester endogenous levels of PI(4,5)P2 at the apical membrane, compromising localization of Sktl and other PI(4,5)P2-binding proteins, and leading to a disrupted morphology of tubular epithelia (not shown).

Taken together, these observations are consistent with a scenario where apically-biased Sktl generates PI(4,5)P2 at the apical membrane of tubular organs. Elevated apical PI(4,5)P2 then serves to reinforce apical targeting of Sktl, and provides an apical cue for additional proteins such as Dia.

## PI(4,5)P2 levels direct apical Dia localization in *Drosophila* tubular organs

We next used Sktl overexpression in *Drosophila* tubular organs to manipulate PI(4,5)P2 levels and determine their capacity to influence Dia localization. The partial apical localization of GFP-Dia-FL in tracheal cells (*Figures 1D and 5A*) is barely detectable at late embryonic stages (*Figure 5C*), but is significantly enhanced upon co-expression with Sktl-RFP (*Figure 5A–D*). Corresponding measurements

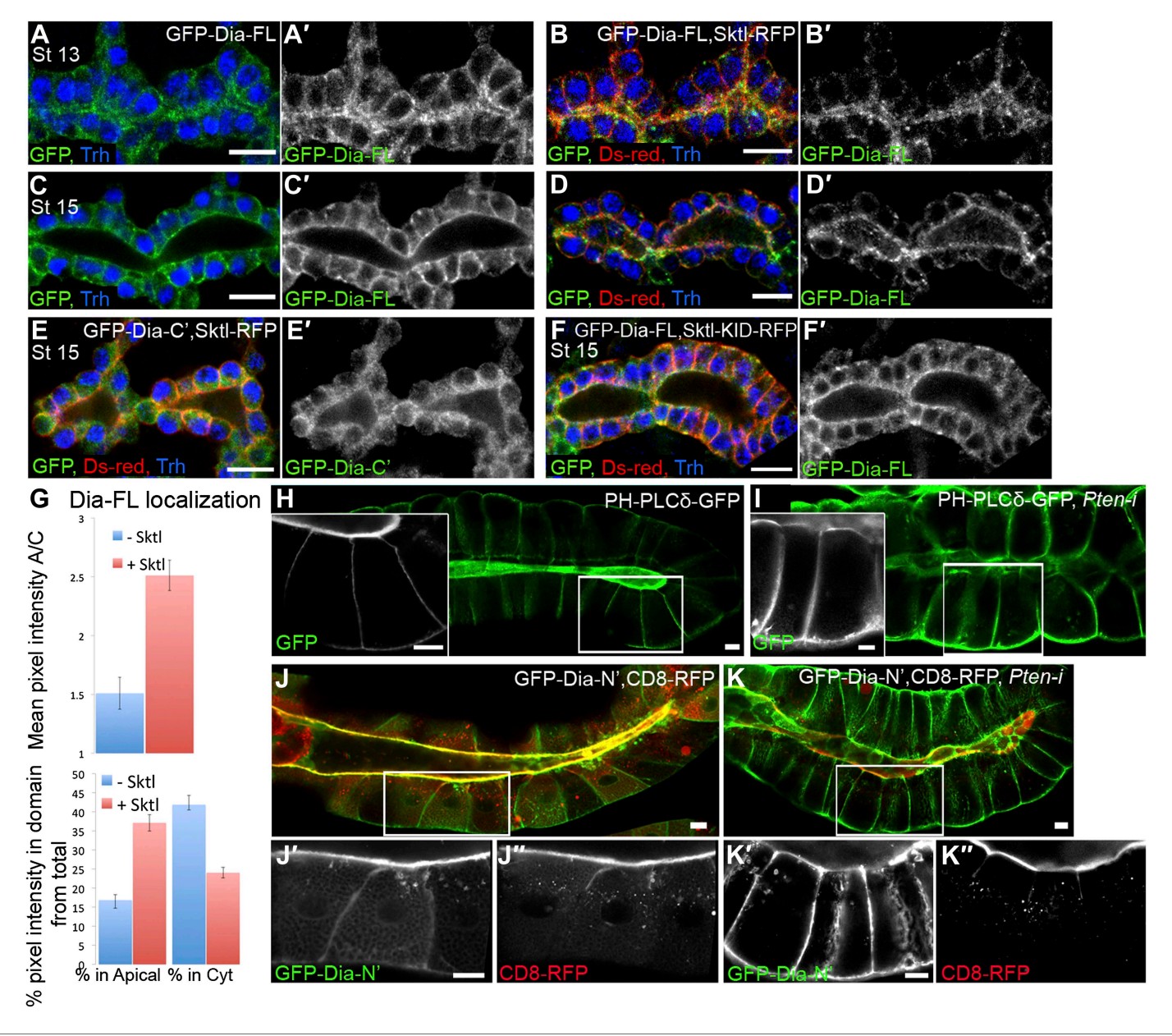

**Figure 5.** PI(4,5)P2 levels regulate Dia apical localization in *Drosophila* tubular epithelia. (**A–F'**) Localization of the indicated constructs was monitored by staining with antibodies against GFP and Ds-Red, following expression in the embryonic trachea under *btl-Gal4*. Primed panels (grey) show the GFP channel alone. Trh (blue) marks tracheal cell nuclei. (**A–D'**) The localization of GFP-Dia-FL is shifted from the cytoplasm to the apical surface following co-expression with Sktl-RFP in stage 13 (**B** and **B'**) and stage 15 (**D** and **D'**) tracheal cells. Compare with the control embryos expressing GFP-Dia-FL alone (**A**,**A'** and **C**,**C'**). (**E**,**E'**) When co-expressed with Sktl-RFP, the localization of GFP-Dia-C' remained entirely cytoplasmic. (**F** and **F'**) Apical enrichment of GFP-Dia-FL is not enhanced upon co-expression with Sktl-KID-RFP. Scale bars, 10 μm. (**G**) Quantification of GFP fluorescence intensity in stage 15 tracheal cells. Upper chart: mean pixel intensity ratio between the apical domain and the cytoplasm. Lower chart: percentage of intensity that is concentrated in the apical and cytoplasmic domains from the total intensity in the cell. Error bars represent standard error. n = 11–20. (**H–K''**) Localization of the indicated constructs was examined by live imaging in third instar larval salivary glands following expression under *fkh-Gal4*. (**H**) In WT cells, the PI(4,5)P2 sensor PH-PLCδ-GFP is highly enriched in the apical membrane. (**I**) Upon co-expression with *Pten*-RNAi, PH-PLCδ-GFP is re-distributed between the cell membranes, and appears cortical. (**J–K''**) Co-expression of GFP-Dia-N' (green) with CD8-RFP (red), which served as a general apical membrane marker. Separate channels (grey) are shown in primed panels. (**J–J''**) In WT cells, both GFP-Dia-N' and CD8-RFP are highly restricted apically. (**K–K''**) In *Pten*-RNAi expressing cells, GFP-Dia-N' is re-distributed between the cell membranes and appears cortical, while CD8-RFP remains restricted to the apical membrane. Scale bars, 20 μm.

of GFP intensities show a clear shift in GFP-Dia-FL distribution from the cytoplasm to the apical domain in Sktl-expressing cells (*Figure 5G*). Conversely, apical enrichment was not observed following co-expression of Sktl with the GFP-Dia-C' construct, which does not contain apical targeting activity (*Figure 5E*). Moreover, GFP-Dia-FL localization was not affected by co-expression with the catalytically inactive form of Sktl (*Figure 5F*). These results suggest that Sktl-dependent increase in PI(4,5)P2 levels at the apical membrane leads to enhanced recruitment of Dia.

An additional regulator of PI(4,5)P2 levels in the membrane is the lipid phosphatase PTEN, which dephosphorylates PI(3,4,5)P3 to form PI(4,5)P2, thus acting as a negative regulator of the PI3K pathway (*Figure 4A*). PTEN has been shown to be associated with the regulation of cell polarity in both *Drosophila* and mammalian systems (*Martin-Belmonte et al., 2007*; *Chartier et al., 2011*). PTEN resides at adherens junctions (*von Stein et al., 2005*), consistent with a role in the maintenance of PI(4,5)P2 levels at the apical membrane. We therefore used RNAi-mediated silencing of *Pten* to manipulate PI(4,5)P2 levels in vivo, and followed the effect on Dia localization. We chose to perform these studies by imaging of live third instar larval salivary glands. This system offers several favorable features, including large polarized cells with clearly distinguishable domains, avoidance of fixation artifacts, and a relatively long period for depletion of gene products through an RNAi-based approach. Monitoring of the GFP-Dia-N' construct expressed in this tissue revealed a highly restricted localization to the apical surface of the cells lining the gland lumen (*Figure 5J*), consistent with our observations in the embryonic salivary gland and other tubular organs.

The PI(4,5)P2 sensor PH-PLCδ-GFP is localized apically in larval salivary glands, similar to other tubular organs (*Figure 5H*). A pronounced redistribution of the sensor was observed following co-expression of *Pten*-RNAi, such that the sensor became evenly localized around the cortex of salivary-gland cells (*Figure 5I*). This redistribution is consistent with studies in MDCK cysts that link the loss of PTEN activity with failure in segregation of PI(4,5)P2 from PI(3,4,5)P3, leading to their homogenous, rather than polarized, membrane distributions (*Martin-Belmonte et al., 2007*).

Similar to the effect on the PI(4,5)P2 sensor localization pattern, expression of a *Pten*-RNAi construct caused a dramatic redistribution of the GFP-Dia-N' construct, from a restricted apical localization to a homogenous distribution throughout the plasma membrane of third instar salivary gland cells (*Figure 5J–K*). To rule out the possibility of secondary effects resulting from a general disruption of cell polarity, we co-expressed an RFP-tagged version of the membrane protein CD8 (CD8-RFP), which resides exclusively in the apical membrane of salivary gland cells (*Xu et al., 2002*). The localization of the CD8-RFP construct remained apically restricted following *Pten* knockdown, indicating that the loss of PTEN specifically affected Dia localization (*Figure 5J″–K″*). Distribution of myristoylated RFP, another apical marker in this tissue, was similarly unaffected (not shown).

In summary, manipulations of PI(4,5)P2 levels profoundly and specifically affect Dia localization in vivo, demonstrating the involvement of PI(4,5)P2 in Dia apical targeting.

## The PI(4,5)P2-Dia interaction is mediated by binding through the BD domain

The construct containing the N-terminal half of Dia (Dia-N') faithfully recapitulated apical targeting (*Figure 2*), thus providing a reporter for the localization of Dia. Which domain within this construct mediates the interaction with PI(4,5)P2? Previous reports showed that the mammalian homologues of Dia, mDia1 and mDia2, directly bind acidic phospholipids via electrostatic interactions, involving basic stretches near the N-terminus of the protein, termed the Basic Domain (BD) (*Ramalingam et al., 2010*; *Gorelik et al., 2011*). Indeed the *Drosophila* Dia N-terminal 60 amino acids are also enriched for basic residues (*Figure 2A*).

To examine a role for this region in apical targeting of *Drosophila* Dia, we generated transgenic flies that contain a variant GFP-Dia-N' reporter lacking its N-terminal basic domain (GFP-Dia-N'ΔBD). The same chromosomal insertion site as the intact GFP-Dia-N' reporter was used, to maintain similar expression levels. Deletion of the BD profoundly reduced apical targeting of the GFP-Dia-N' reporter in larval salivary glands, as well as in other tubular tissues (*Figure 6A–B, 7G* and *Figure 6—figure supplement 1*), demonstrating its significance for Dia localization.

To assess the apical targeting capacity of the BD on its own, a GFP-Dia-BD construct (*Figure 2A*) was expressed in larval salivary glands and other tubular tissues. Remarkably, no apical localization bias was observed for this protein, as was also confirmed by comparison to the localization of GFP alone,

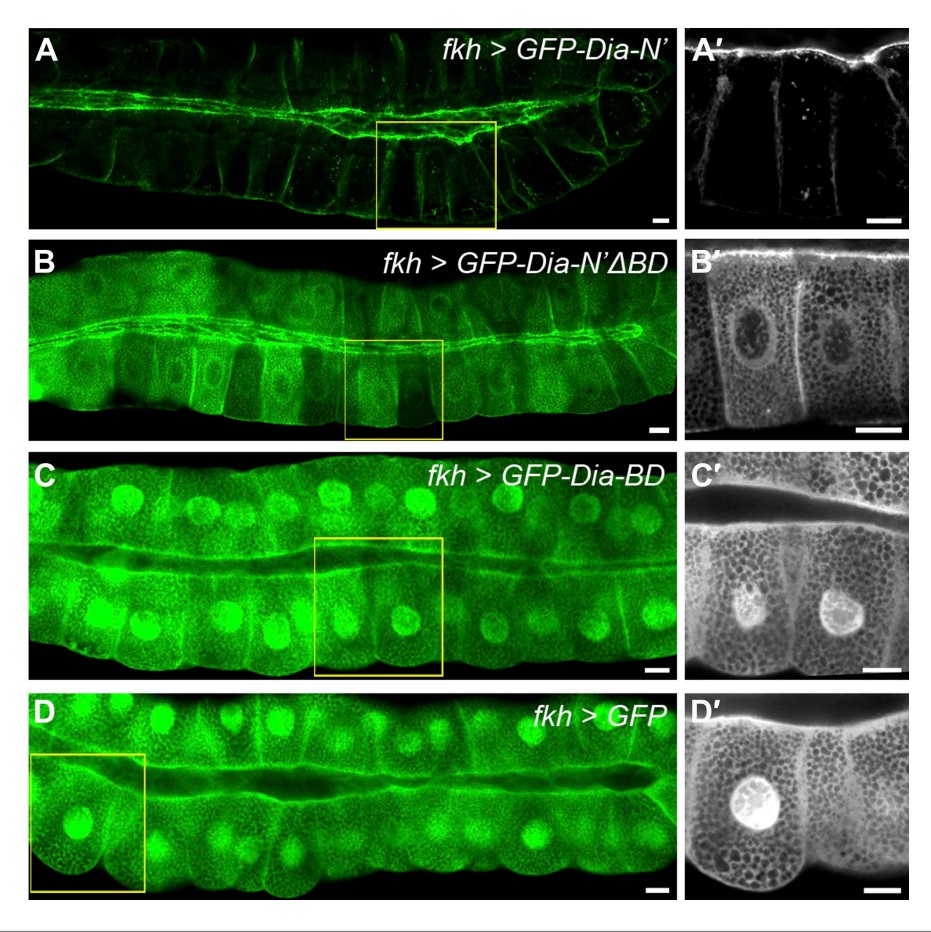

**Figure 6**. PI(4,5)P2-Dia interaction is mediated by direct binding through the BD domain. Live imaging of GFP-tagged Dia constructs (*Figure 2A*), expressed in third instar larval salivary glands by *fkh-Gal4*. Primed panels (grey) show an enlargement of the marked area. (**A**–**B′**) Apical localization of GFP-Dia-N′ΔBD (**B** and **B′**) is significantly weaker than GFP-Dia-N′ (**A** and **A′**). (**C**–**D′**) GFP-Dia-BD displays a cytoplasmic and nuclear distribution (**C** and **C′**), which is comparable to that of GFP alone (**D** and **D′**). The apparent enrichment in the apical domain, resulting from absence of granules in this area, as well as nuclear localization, is observed with both constructs. Scale bars, 20 µm. See also *Figure 6—figure supplement 1*.

The following figure supplements are available for figure 6:

**Figure supplement 1**. Dia constructs show similar localization patterns in all tubular organs.

and measurements of GFP apical enrichment (*Figure 6C–D* and *Figure 6—figure supplement 1*). Thus, despite being critical for Dia localization, the basic domain is not sufficient on its own to drive apical targeting.

## Direct binding to Rho1 anchors Dia to the apical surface

The inability of the BD to drive apical targeting on its own, suggested that PI(4,5)P2 may function in combination with additional factors to target Dia apically. The N-terminus of Dia is composed of several distinct domains that can serve as targets for apical recruitment (*Figure 2A*). Prominent among these is the GTPase-binding domain, through which Rho1 binds and activates the actin nucleation and elongation capacities of Dia (*Otomo et al., 2005*; *Rose et al., 2005*). An initial indication that Rho1 can serve as an apical recruitment factor for Dia came from examination of Rho1 protein localization. A GFP-Rho1 construct specifically expressed in third instar larval salivary gland cells showed clear enrichment at the apical surface (*Figure 7A*), as did a GFP fusion protein expressed from the endogenous Rho1 promoter (not shown). Apical enrichment of GFP-Rho1 was not altered upon co-expression with

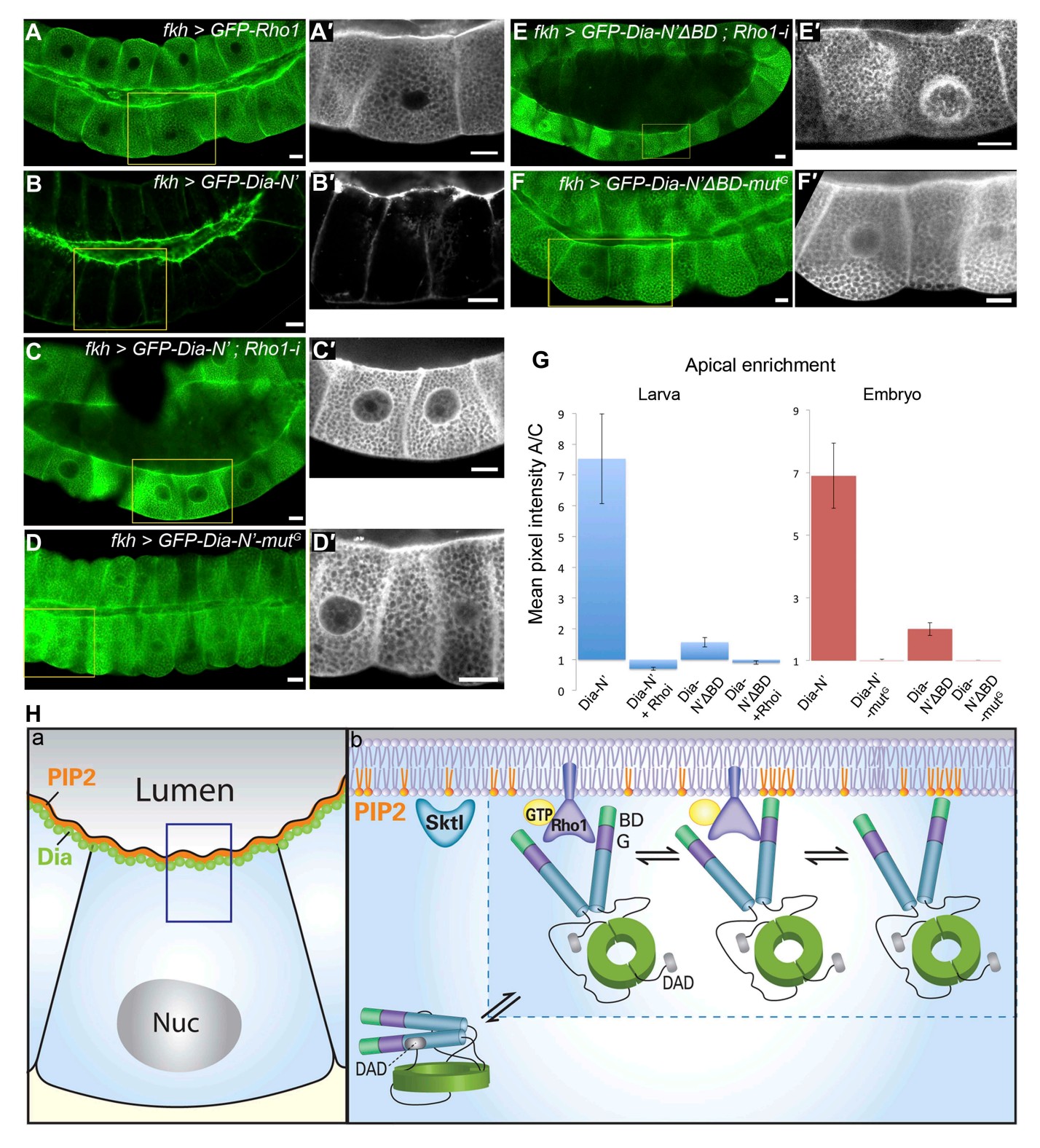

**Figure 7**. Direct binding to Rho1 mediates Dia apical localization. (**A–F′**) Live imaging of GFP-tagged constructs (**Figure 2A**) expressed in third instar larval salivary glands under **fkh-Gal4**. Primed panels (grey) show an enlargement of the marked area. (**A** and **A′**) GFP-Rho1 is distributed between the apical domain and the cytoplasm. (**B–D′**) Apical localization of Dia-N′ is Rho1 dependent. Co-expression with Rho1-RNAi dramatically alters GFP-Dia-N′ localization from apical restriction (**B** and **B′**) to a complete cytoplasmic localization (**C** and **C′**). Correspondingly, GFP-Dia-N′-mutG, a variant lacking

Figure 7. Continued on next page

*Figure 7. Continued*

Rho-1 binding capacity, assumes an entirely cytoplasmic distribution (**D** and **D'**). Note the expanded lumen of Rho1-RNAi expressing salivary glands (**C** and **C'**), previously reported for Rho1 mutant alleles (***Xu et al., 2011***), indicative of the loss of active Rho1 in this system. (**E–F'**) Apical localization of Dia-N'ΔBD is Rho1 dependent. Co-expression with Rho1-RNAi dramatically alters GFP-Dia-N'ΔBD localization to a cytoplasmic localization (**E** and **E'**, compare with ***Figure 6B***), while GFP-Dia- N'ΔBD -mutG, lacking Rho-1 binding capacity, assumes an entirely cytoplasmic distribution (**F** and **F'**). Scale bars, 20 μm. (**G**) Quantification of GFP intensity in larval (left chart) and embryonic (right chart) salivary gland cells, expressing the indicated constructs under *fkh-Gal4* (***Figures 6 and 7***, ***Figure 6—figure supplement 1*** and ***Figure 7—figure supplement 2***). Apical enrichment is represented by the mean pixel intensity ratio between the apical domain and the cytoplasm. For larval salivary gland cells, the values are normalized to measurements of cells expressing cytoplasmic GFP. Due to morphological changes resulting from Rho1–RNAi expression, Dia apical localization is even lower than the baseline. Error bars represent standard error. n = 10–24 glands. (**H**) A model describing the dynamics of Dia targeting to the apical membrane in *Drosophila* tubular organs. a) Enrichment of both PI(4,5)P2 and Dia at the apical membrane is an inherent feature of *Drosophila* tubular epithelial cells. b) In a closed conformation, Dia assumes a cytoplasmic distribution. Dia binding to Rho1-GTP generates an initial apical bias, and promotes PI(4,5)P2 binding via exposure of the N-terminal region and enrichment near the apical membrane. The Dia- PI(4,5)P2 interaction further stabilizes apical membrane association. Dia can shift dynamically between three states, being bound to each of the apical cues alone, or to both simultaneously, increasing the probability for re-binding and therefore the efficiency of apical targeting. See also ***Figure 7—figure supplement 1–3***.

The following figure supplements are available for figure 7:

**Figure supplement 1**. Apical enrichment of Rho1 is not dependent on Dia.

**Figure supplement 2**. Rho1 binding is critical for Dia localization in embryonic tubular organs.

**Figure supplement 3**. Rho1 and PI(4,5)P2 do not regulate each other's localization pattern.

a *dia*-RNAi construct, indicating that it is not mediated by Dia protein or by actin structures generated by Dia (***Figure 7—figure supplement 1***).

We next examined the localization pattern of GFP-Dia-N' following depletion of Rho1. The ability to follow the inherently open Dia-N' construct, allowed to critically examine the role of Rho1 as a physical anchor for Dia, independent of its role as an activator. GFP-Dia-N' localization was dramatically altered upon co-expression with a *Rho1*–RNAi construct, and displayed a uniform distribution in the cytoplasm (***Figure 7B–C***). Furthermore, a similar outcome was obtained following expression of GFP-Dia-N'-mut[G] (***Figure 2A***), a GFP-Dia-N' variant, which is mutated at residues that confer Rho1 binding (***Otomo et al., 2005***; ***Rose et al., 2005***; ***Lammers et al., 2008***). The distribution of GFP-Dia-N'-mut[G] was essentially cytoplasmic upon expression in larval salivary glands, as well as in various embryonic tubular tissues (***Figure 7D*** and ***Figure 7—figure supplement 2***), indicating that direct interaction with Rho1 mediates Dia targeting. Thus, apically-localized Rho1 functions to physically anchor Dia to the apical surface, in addition to opening the protein. We assume that Rho1 exerts both activities only in the active, GTP-bound form (***Rose et al., 2005***).

We have shown that binding to both Rho1 and PI(4,5)P2 contribute to apical targeting of Dia. We next wanted to explore the possible relationship between these two cues. First, Rho1 localization was not altered following different manipulations designed to modify PI(4,5)P2 levels and affect Dia localization, including Sktl over-expression and PTEN depletion (***Figure 7—figure supplement 3***). Similarly, when we monitored the PI(4,5)P2 reporter PH-PLCδ-GFP, it remained apically enriched following co-expression with a *Rho1*–RNAi construct (***Figure 7—figure supplement 3***), indicating that Rho1 does not directly regulate PI(4,5)P2 distribution.

Second, we analyzed the inter-relationship between the two cues with respect to Dia apical localization. Previous studies of the mDia1–Rho1 interaction showed that the BD is dispensable for the formation of an mDia1–Rho1 complex (***Otomo et al., 2005***; ***Rose et al., 2005***). Co-expression of GFP-Dia-N'ΔBD with *Rho1*-RNAi or expression of a GFP-Dia-N'ΔBD mut[G] construct resulted in complete loss of apical localization (***Figure 7E–G***). This indicates that the residual apical localization observed for the GFP-Dia-N'ΔBD construct (***Figure 6B***) can be attributed to Rho1 binding, consistent with the notion of two distinct functional modules for Dia binding. However, the inability of the BD to direct even a mild bias of apical localization on its own (***Figure 6C***), suggests that the Dia–PI(4,5)P2 interaction requires simultaneous binding of an apical cue through an additional domain.

Finally, quantification of Dia apical enrichment in larval salivary gland cells revealed that the ultimate apical distribution of Dia is strongly affected by the combined influence of both cues. While the

capacity to bind PI(4,5)P2 does not promote apical localization in the absence of Rho1, it strongly enhances apical targeting of Dia when Rho1 is present (*Figure 7G*, compare Dia-N'ΔBD with Dia-N'). Similar results were obtained in embryonic salivary glands, where a construct defective in PI(4,5)P2 binding (Dia-N'ΔBD) showed weak apical enrichment (≈2), while a construct defective in Rho1 binding (Dia-N'ΔBD-mut[G]) showed no apical bias whatsoever (*Figure 7G*). However, the Dia-N' construct, in which the binding domains for both cues are intact, displayed apical enrichment values of ≈7, clearly indicative of synergy between Rho1 and PI(4,5)P2 in promoting Dia apical localization. Taken together, we show that a partial apical localization bias of Dia by Rho1 binding is significantly enhanced by binding to PI(4,5)P2.

## Discussion

The ability to recruit Dia, one of the central nucleators of actin microfilaments, to the cell membrane, is important for its capacity to trigger processes of membrane extension and intracellular trafficking. We focused our study of Dia recruitment on tubular epithelial tissues, where Dia was shown to be specifically localized to the apical membrane. This tight apical localization is critical for the formation of polarized actin tracks, guiding apical movement of secretory vesicles that are associated with MyosinV motors (*Massarwa et al., 2009*). Mechanisms that regulate the apical localization of Dia were examined in distinct tubular organs such as the trachea, salivary glands and hindgut, and at different stages of *Drosophila* development. This experimental flexibility allowed us to utilize the specific advantages of different tissues, and identify features which are universal to all tubular organs. We found that apical recruitment of Dia is mediated by binding to two distinct elements, PI(4,5)P2 and Rho1. Further analysis suggests that synergism between these two partial signals ensures highly efficient apical localization of active Dia.

### PI(4,5)P2-Dia interaction serves as a cue for apical targeting

The enrichment of PI(4,5)P2 in apical membranes of tubular organs (*Martin-Belmonte et al., 2007*) suggested that PI(4,5)P2-binding could serve as a common localization cue for Dia, shared by different tubular organs. Indeed, a reporter for the endogenous levels of PI(4,5)P2 displayed a highly significant apical enrichment in *Drosophila* tubular organs (*Figure 4*). The causal role of PI(4,5)P2 in recruiting Dia was demonstrated in several ways. First, *Drosophila* Dia was apically localized in the heterologous MDCK cell culture system, and its localization was shifted basally within minutes of adding PI(4,5)P2 to the basal membranes (*Figure 3*). We wish to point out that while in this system Dia localization is regulated by PI(4,5)P2 under both excess and normal conditions, it is currently unknown whether an interaction between Dia and mammalian Rho GTPases also contributes to the apical localization. Second, elevation in *Drosophila* tubular organs in the levels of Sktl, a PIP5-kinase that generates PI(4,5)P2, gave rise to a corresponding enhancement in the level of apical Dia. Conversely, knockdown of the *Pten* gene encoding the PI(3,4,5)P3 phosphatase in these tissues resulted in redistribution of PI(4,5)P2, consistent with previous observations (*Martin-Belmonte et al., 2007*). A corresponding alteration in Dia localization further supports the causal role of PI(4,5)P2 in its localization (*Figure 5*).

It is interesting to note that generation of PI(4,5)P2 at the apical membrane by Sktl exhibits self-amplifying properties. Over-expression of tagged Sktl resulted in both a massive increase in the levels of apical PI(4,5)P2 (not shown), and accumulation of Sktl at the same domain. In contrast, a Sktl protein that is catalytically inactive was distributed uniformly, when expressed in a similar manner (*Figure 4*). We assume that the signal targeting Sktl apically may be limited, and capable of providing only an initial bias. The resulting enrichment of PI(4,5)P2 following the activity of Sktl can lead to the recruitment of additional Sktl molecules and amplify this apical bias.

The link between phospholipids and localization of formins extends to the plant kingdom. In the moss *Physcomitrella patens,* class-II formins bind PI(3,5)P2, leading to their cortical localization, and allowing them to mediate polarized growth (*van Gisbergen et al., 2012*). Taken together, phospholipids are emerging as important regulators of formin localization and activity at the tissue and organismal level.

The mammalian formins mDia1 and mDia2 bind phospholipids through a basic domain (BD), located at the extreme N-terminus. This BD was shown to regulate both activity and recruitment to membranes in cell culture models (*Ramalingam et al., 2010*; *Gorelik et al., 2011*). Similar to the mammalian homologues, *Drosophila* Dia also contains an N-terminal basic domain which is critical for apical localization (*Figure 6*), and is likely to mediate the association with PI(4,5)P2 through electrostatic

interactions with the acidic phospholipid. These interactions are known to be weak, and are usually insufficient to stably associate proteins with the membrane under physiological conditions, requiring coupling to additional membrane-targeting motifs (*Mulgrew-Nesbitt et al., 2006*). Indeed, the Dia BD proved insufficient to direct apical targeting on its own (*Figure 6*). In this respect, Dia differs from plant formins, which commonly contain a structured PTEN domain for phospholipid binding. This domain confers strong and more specific association, and is consequently sufficient for cortical localization, without requirement for additional targeting factors (*van Gisbergen et al., 2012*).

## Binding to Rho1 activates and anchors Dia to the apical membrane

Insufficiency of the BD in mediating Dia apical targeting led us to search for additional targeting signals. The constitutively activated, open conformation of Dia was significantly more accessible to apical targeting (*Figure 1*), probably as a result of exposing the N-terminus more readily to apical-targeting factors. Interestingly, even high levels of the open conformation of Dia were efficiently targeted, indicating that the targeting machinery is in relative excess. Normally, the generation of the open Dia conformation is facilitated by binding to Rho1 (*Otomo et al., 2005*; *Rose et al., 2005*). In addition, for some formin-family proteins, phosphorylation of residues adjacent to the DAD domain by a Rho1 effector, ROCK-1, is an essential element in the activation process (*Takeya et al., 2008*; *Staus et al., 2011*). It remains to be determined whether ROCK-1 also plays a role in activation of Dia in *Drosophila* tubular organs.

Rho1 also contributes directly to apical-membrane anchoring of Dia. We were able to deplete Rho1 in larval salivary glands, and observed a complete loss of apical targeting (*Figure 7*). The ability to follow localization of an open form of Dia allowed us to separate the roles of Rho1 in Dia activation and localization. The open form of Dia still requires binding to Rho1 for its apical localization, as was shown by elimination of Rho1 and by mutating the Rho1-binding domain of Dia (*Figure 7*).

There are indications for apical enrichment of Rho1 protein in *Drosophila* tubular organs (*Massarwa et al., 2009* and *Figure 7*). The distribution of the activated form of Rho1 may be even more apically biased. It appears, however, that Rho1 requires different signals than Dia for its apical localization. Alterations in the PI(4,5)P2 distribution, which resulted in dramatic effects on the localization of Dia, had only a marginal effect on the localization of Rho1 (*Figure 7—figure supplement 3*). While Dia localization depends on Rho1, the localization of Rho1 is independent of Dia and the apical actin cables it forms (*Figure 7—figure supplement 1*), suggesting a hierarchy rather than a reinforcing feedback loop between the two proteins. One possibility is that apical Rho1 localization involves anchoring to Rho-GEFs, the specific guanine-exchange factors that activate Rho1, and were also shown to be enriched in apical membranes of *Drosophila* tubular organs (*Massarwa et al., 2009*).

## Cooperating interactions target Dia apically

Our results demonstrate that the tight apical localization of Dia is achieved by combining functionally distinct apical-biasing cues. Neither Rho1 nor PI(4,5)P2 is sufficient to efficiently recruit Dia apically on their own. Nevertheless, combining these two relatively weak interactions gives rise to dramatic apical localization of Dia, demonstrating a synergistic contribution (*Figure 7*). The cooperative contribution of multiple domains within the same protein to drive membrane targeting is termed 'Coincidence detection' (*Lemmon, 2008*). This mechanism for localization of a multidomain protein can be used to generate enhanced targeting specificity. While both Rho1 and PI(4,5)P2 may separately reside in other subcellular domains, Dia will bind with high avidity exclusively to the apical membrane that encompasses both cues.

The modular nature of these targeting mechanisms allows to examine the contribution of each cue, and the significance of their cooperation in Dia apical restriction. Binding to Rho1 alone was weak but nevertheless detectable. Conversely, binding to PI(4,5)P2 alone was not capable of generating any apical localization bias, despite its marked contribution to apical recruitment in the presence of Rho1 (*Figure 7*). This asymmetry between the two cues suggests a model for the dynamics of Dia-apical targeting (*Figure 7H*). Binding to activated Rho1 may represent the more prevalent scenario for the initial interaction of Dia with the apical membrane. This interaction will also give rise to an open conformation of Dia, which is more amenable to targeting, and will provide physical proximity to the membrane. Membrane proximity can now allow the interaction of Dia with PI(4,5)P2, which was not favored when the Dia protein was cytoplasmic, due to low affinity restrictions. The Dia-PI(4,5)P2 interaction stabilizes association with the apical membrane, resulting in synergy in the combined action of Rho1

and PI(4,5)P2 in Dia apical localization. Once in the vicinity of the apical membrane, Dia can shift dynamically among three states, being bound to each of the cues alone, or to both simultaneously. This feature can increase the effective levels of apical-targeting cues, since they do not necessarily have to be used simultaneously at all times.

Mechanisms that similarly utilize the synergistic activity of two proteins may be used in the regulation of other actin regulators, which integrate between multiple signals. The activity of N-WASP, the activator of the ARP2/3 actin nucleation complex, was shown to be dependent on the synergistic binding of the Rho GTPase Cdc42 and of PI(4,5)P2 to distinct domains in the protein. However, this binding is not reported to lead to targeting of the protein to distinct subcellular domains, but rather to localized activation (*Prehoda et al., 2000*; *Padrick and Rosen, 2010*).

While the apical targeting of Dia protein is very prominent, additional regulation operating on other tiers of the pathway may ensure enhanced fidelity of the localization. Although these mechanisms seem redundant, they may provide higher accuracy to the system. As previously noted, *dia* mRNA is also apically localized by a mechanism that is distinct from the protein localization machinery, leading to localized translation of Dia protein at the apical membrane (*Massarwa et al., 2009*). Indeed, we observed a more restricted apical distribution of endogenous Dia compared with the over-expressed protein, which is not localized at the RNA level (*Figure 1*). In addition, a bias in the apical localization of Dia activators (Rho-GEFs and Rho1) may provide another level of refinement. Finally, the bias in Rho-GEFs may be influenced by apical PI(4,5)P2 levels, providing an additional layer of convergence between distinct signals (*Murray et al., 2012*; *Viaud et al., 2012*).

It is likely that this general mechanism in *Drosophila* tubular organs extends to tubular tissues in the mammalian system. A collaboration between PI(4,5)P2 and a Rho-family GTPase in membrane recruitment of mammalian mDia2 was observed in a cell culture system (*Gorelik et al., 2011*), and here we have demonstrated the capacity of PI(4,5)P2 to influence Dia targeting in mammalian MDCK cysts (*Figure 3*). Furthermore, the link between activation and localization, as well as the identity of the critical domain for localization of formin family proteins, appear to be conserved (*Carramusa et al., 2007*; *Watanabe et al., 2010*). In this context, we have recently examined the tubular epithelium of secretory acinar cells in the mouse pancreas, and shown that activated mDia1 is targeted to the apical membrane (*Geron et al., 2013*). Thus, the Dia apical-targeting machinery appears to be common to a variety of tubular tissues, regardless of their origin or physiological function.

In conclusion, this work uncovers a mechanism by which Dia is targeted to the apical membrane of tubular epithelia, thereby restricting actin cable formation, and consequently secretion, to a distinct membrane domain of these cells. This localization mechanism is likely to be universal to tubular epithelia in a broad range of organisms.

## Materials and methods

### DNA constructs

For the generation of GFP-Dia-N', GFP-Dia-N'ΔBD and GFP-Dia-BD, eGFP was cloned into pUAST–attB at the BglII–EcoRI sites. The relevant segments from a Dia cDNA (*Figure 2A*) were then cloned using the NotI-XbaI sites. GFP-Dia-N'-mut[G] and GFP-Dia-N'ΔBD-mut[G] were created by site directed mutagenesis (Phusion, NEB), replacing nucleotides TG (440–441) with AC and AGC (451–453) with GAG, using GFP-Dia-N' and GFP-Dia-N'ΔBD, respectively, as templates. The resulting constructs were subsequently cloned into GFP-pUAST-attB at the NotI-XbaI sites. All constructs were sequenced, and injected into attP18 lines (*Markstein et al., 2008*). For expression in MDCK cysts, Dia-FL and Dia-ΔDAD were cloned into pEGFP-C1 by inserting the appropriate segments from a Dia cDNA using the SalI and HindIII sites.

### Fly strains

The following lines were used: UAS-GFP-Dia-FL, UAS-Dia-ΔDAD, UAS-Dia-C', UAS-Dia-DD-C' (referred to as Dia, FH1-FH2 and DDFH1FH2, respectively in *Homem and Peifer, 2009*). UAS-GFP-Dia-N', UAS-GFP-Dia-N'ΔBD, UAS-GFP-Dia-BD, UAS-GFP-Dia-N'-mut[G] and UAS-GFP-Dia-N'ΔBD-mut[G] were generated by standard phi31 germline transformation procedures. Additional strains included UAS-PH-PLCδ-GFP (*von Stein et al., 2005*), UAS-Sktl-RFP, UAS-Sktl-KID-RFP (*Raghu et al., 2009*), UAS-GFP (BM 5431), UAS-*Pten-RNAi* (VDRC 101475), UAS-*Rho1-RNAi* (VDRC 12734), UAS-*dia-RNAi* (VDRC 20518), UAS-GFP-Rho1 (BM 9392), UAS-CD8-RFP (BM 32218), hsFLP (BM 6), P{*Gal4-ActRc*(FRT.CD2).P}

S,P{UAS-RFP.W}3/TM3 (BM30558). Gal4 drivers were *btl-Gal4* (*breathless*, expresses Gal4 in tracheal cells), *drm-Gal4* (*drumstick*, expresses Gal4 in embryonic proventriculus, anterior midgut, posterior midgut, Malpighian tubules, hindgut and salivary glands) and *fkh-Gal4* (*fork head*, expresses Gal4 in the salivary gland).

## Immunohistochemistry

Standard embryo fixation and staining procedures were followed (*Patel, 1994*). Detection of Dia was carried out according to (*Massarwa et al., 2009*).

Primary antibodies used were anti Dia (rabbit 1:250 [*Grosshans et al., 2005*]), Crumbs, Dlg (mouse 1:100; DSHB, University of Iowa, USA), GFP (chicken 1:500; Abcam), GFP (rabbit 1:700; Life Technologies, for MDCK cysts), Trh (rat 1:100), FasIII (mouse 1:20), gp135 (Podocalyxin, mouse 1:2000; from G Ojakian), DsRed (Rabbit 1:500; Clontech), Actin (mouse 1:1000, Sigma). Anti-mouse, rabbit and chicken Alexa-conjugated secondary antibodies were obtained from Invitrogen and diluted 1:800. Anti-rat Cy5-conjucated secondary antibody was purchased from Jackson ImmunoResearch and diluted 1:200. Nuclei of MDCK cells were counterstained with Hoechst (Molecular Probes).

## Image acquisition and analysis

Live embryos and dissected third instar larval salivary glands were imaged following mounting in Halocarbon oil 700 (Sigma). Images of immunofluorescent and live samples were acquired using a Zeiss LSM 710 confocal scanning system, using ×20 N.A 0.5 or ×63 N.A 1.4 objectives, and processed using Adobe Photoshop CS3. For quantification of apical localization, mid-plane non-saturated images were selected. Mean intensities of GFP fluorescence were measured by ImageJ. The different subcellular domains were demarcated by hand according to specific stainings.

## Stable MDCK line generation, tissue culture and manipulation

3D culture, immunofluorescence staining and microscopy of MDCK cysts, as well as lipid delivery to the cysts were performed as previously described (*Martin-Belmonte et al., 2007*).

### Stable cell line generation and tissue culture

MDCK cells were maintained in minimal essential medium (MEM) containing Earles' balanced salt solution (Life Technologies) supplemented with 5% fetal calf serum, 100 units/ml penicillin and 100 µg/ml streptomycin in a 5% $CO_2$ humidified chamber. For MDCK cells stable transfection, cells were selected for 2–3 weeks in Geneticin (800 µg/ml) and FACS was performed to enrich the GFP-positive cell population (FACS Aria).

### 3D culture of MDCK cysts

Cells were trypsinised and triturated to a single cell suspension at $2 \times 10^4$ cells/ml. Cells were plated onto a thin, pre-solidified layer of basement membrane matrix (Matrigel; BD Bioscience) and grown for 3–4 days in cell culture media containing 2% matrigel.

### Lipid delivery to cysts

The histone:phospholipid complexes were prepared by incubating 300 µM of PI(4,5)P2 and 100 µM of histone for 5–10 min at room temperature. Cysts were pre-treated with trypsin (no EDTA, 10 min) prior to addition of the lipid:histone complexes. Immediately prior to addition to cysts the complexes were diluted in pre-warmed HBSS containing 1% serum and 2% matrigel then incubated for the indicated time in a 37°C/5% $CO_2$ humidified chamber.

### Immunofluorescence microscopy

Samples were examined and imaged using a Zeiss 510 LSM confocal microscope fitted with a C-Apochromat 63x/1.2NA W objective.

## Acknowledgements

We thank the following scientists and organizations for generously providing fly strains and antibodies: JA Brill, J Großhans, G Ojakian, M Peifer, P Raghu, the Vienna *Drosophila* RNAi Center, the Developmental Studies Hybridoma Bank, and the Bloomington *Drosophila* Stock Center. We thank G Schreiber and R Zaidel-Bar for helpful discussions, R Massarwa for critical reading of the manuscript, and all members of the Shilo laboratory for helpful discussions. BZS is an incumbent of the Hilda and Cecil Lewis chair for Molecular Genetics.

## Additional information

### Funding

| Funder | Grant reference number | Author |
| --- | --- | --- |
| US-Israel Binational Science Foundation | 2009186 | Keith E Mostov, Eyal D Schejter, Ben-Zion Shilo |
| National Institutes of Health | DK074398, DK091530 | Keith E Mostov |
| CJ Martin Fellowship, National Health and Medical Research Council, Australia | | Annette M Shewan |
| Schoenheimer Foundation | | Ben-Zion Shilo |

The funders had no role in study design, data collection and interpretation, or the decision to submit the work for publication.

### Author contributions

TR, Conception and design, Acquisition of data, Analysis and interpretation of data, Drafting or revising the article; AMS, Acquisition of data, Analysis and interpretation of data; KEM, Conception and design, Analysis and interpretation of data; EDS, B-ZS, Conception and design, Analysis and interpretation of data, Drafting or revising the article

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
