## [Decision Letter]

Thank you for submitting your interesting manuscript “Apical targeting of the formin Diaphanous in *Drosophila* tubular epithelia” to *eLife*. We are happy to say that all the reviewers were generally positive about your submission, and feel that the quality and scope is appropriate for *eLife*: “The experiments described here show the power of *Drosophila* genetics to reveal interdependent regulatory mechanisms that are key to protein localization (and presumably function) in tissues of a multicellular organism. Overall, the manuscript is beautifully written, the experiments are clear, and clearly described, and the conclusions are generally well justified by the data.”

However, there were some issues that should be addressed in your revised submission. All the reviewers felt that the manuscript would benefit from additional quantification of the localization of several constructs, as well as some additional verification of the biochemical interactions indicated below.

1) Quantification of BD localization: “My one significant concern relates to Figure 6 and the authors’ assertion that the Dia-BD domain is not sufficient for apical localization, an assertion that then leads to the model presented in Figure 8 and in the text. Although I understand that the authors’ believe that there is an inherent bias toward apical staining in the larval salivary gland, the image shown in Figure 6 clearly seems to indicate an apical localization for this construct. For this reason it is difficult to accept the authors’ assertion that this construct shows no apical enhancement. I suggest that the authors need to carefully, and quantitatively, compare the localizations of GFP alone (displayed in a very small panel in Figure 4) to that of the GFP-Dia-BD construct to validate their claim that the two are indistinguishable. This can be done using the same sort of quantitative analysis that they used for Figures 1, 2, 5 and 7.”

2) Quantification of Rho1 localization in the figure supplements: “In the supplemental figures, it is stated that disruptions of PI(4,5)P2 did not affect Rho1, and that Rho1 localization is independent of Dia. In the current quantitation, the authors compared apical to cytoplasmic Rho1; however, it appears that the lateral membrane localization of Rho1 increases when Skittles is expressed. It also seems as if the localization of Rho1 changes upon PTEN RNAi. Quantitation of apical membrane compared to basolateral membranes is needed to clarify these points.”

3) Biochemical validation of the Rho1 mutation (Figure 7): “to complete the biochemical connection, as well as showing that the interaction between Rho1 and Dia is not affected by removal of the BD domain, it is also necessary to do the pull-downs with the mut^G^ version used in vivo to demonstrate that it does not bind Rho1.

Finally, the reviewers had several suggestions that they thought would improve the manuscript, though these changes are not essential for acceptance:

A) The manuscript would be improved by the addition of immunoblotting to test the relative levels of FL versus ΔDAD.

B) The claim that the PIP2-Dia interaction is mediated by direct binding through the BD domain would be better substantiated by lipid flotation assays showing that the BD domain binds PIP2 and that the introduced point mutations abrogate binding. As these experiments take time and may not be possible in a reasonable time frame, the authors should at least acknowledge that direct binding was not tested.

C) The results with exogenous PIP2 and MDCK cells strongly suggest that PIP2 alone is sufficient to direct Dia localization, if one assumes that there was no effect on RhoA localization in this experiment. Ideally RhoA localization should be examined in the MDCK assay; however, at the very least the authors should reconcile this result with their model, which suggests that even in the open conformation Dia requires interaction with Rho to localize properly.

---

## [Author Response]

*1) Quantification of BD localization: “My one significant concern relates to Figure 6 and the authors’ assertion that the Dia-BD domain is not sufficient for apical localization, an assertion that then leads to the model presented in Figure 8 and in the text. Although I understand that the authors’ believe that there is an inherent bias toward apical staining in the larval salivary gland, the image shown in Figure 6 clearly seems to indicate an apical localization for this construct. For this reason it is difficult to accept the authors’ assertion that this construct shows no apical enhancement. I suggest that the authors need to carefully, and quantitatively, compare the localizations of GFP alone (displayed in a very small panel in Figure 4) to that of the GFP-Dia-BD construct to validate their claim that the two are indistinguishable. This can be done using the same sort of quantitative analysis that they used for Figures 1, 2, 5 and 7.*”

We now show that the localizations of the constructs GFP-Dia-BD and GFP alone are indistinguishable in several ways:

•We added an image of the GFP-alone construct next to the GFP-Dia-BD construct in Figure 6, instead of the small image in Figure 4. The apical bias that results from the unique morphology of these cells is apparent with both constructs.

•Quantitative analysis of apical localization comparing the two constructs in both larval and embryonic salivary gland cells was added in a new supplementary figure, Figure 6—figure supplement 1. This analysis clearly shows that the values for the two constructs are similar.

Images of these constructs as well as of GFP-Dia-N’ and GFP-Dia-N’ΔBD in embryonic tissues were transferred to this figure from Figure 2—figure supplement 1.

*2) Quantification of Rho1 localization in the figure supplements: “In the supplemental figures, it is stated that disruptions of PI(4,5)P2 did not affect Rho1, and that Rho1 localization is independent of Dia. In the current quantitation, the authors compared apical to cytoplasmic Rho1; however, it appears that the lateral membrane localization of Rho1 increases when Skittles is expressed. It also seems as if the localization of Rho1 changes upon PTEN RNAi. Quantitation of apical membrane compared to basolateral membranes is needed to clarify these points.*”

Quantification of Rho1 localization in the lateral membrane upon Sktl overexpression was added to Figure 7—figure supplement 3. The ratio between the apical membrane and the lateral membrane (left chart, A/SJ, measured in the septate junction domain) even slightly increases upon Sktl over-expression, indicating that there was no increase in the lateral localization of Rho1. The percentage of intensity that is concentrated in the SJ domain (right chart) is not significantly changed upon Sktl overexpression.

Quantification of Rho1 localization at the basal membrane upon PTEN-RNAi expression was added to Figure 7—figure supplement 3. A very mild decrease in the ratio between the apical and basal domains (A/B) is observed.

*3) Biochemical validation of the Rho1 mutation (Figure 7): “to complete the biochemical connection, as well as showing that the interaction between Rho1 and Dia is not affected by removal of the BD domain, it is also necessary to do the pull-downs with the mut^G^ version used in vivo to demonstrate that it does not bind Rho1*.

We have carried out the co-IP studies of Dia constructs with Rho1, following expression in larval salivary glands. While GFP alone did not pull down Rho1, all other constructs (including the mut^G^) displayed a band of equal intensity that was recognized by Rho antibodies. We do not believe this reflects a bona-fide interaction, and have therefore removed this experiment from the paper. We also went one step further and carried out mass spectrometry analysis on a similar pull down with Dia-DAD (using a larger number of salivary glands, up to 500 per sample). The Rho1 protein could not be identified by this analysis, indicating that the sensitivity of the method does not allow the detection of endogenous levels of activated Rho that bind Dia, or that the extraction conditions used in the pull down were not sufficiently harsh to effectively remove membrane embedded proteins like Rho1.

Nevertheless, based on previous reports (31; 37; 19), we are confident that the two mutations we have introduced in mut^G^ (in two residues conserved between Dia and mDia1 and mDia2) lead to abrogation of binding to Rho1.

There are a number of indications from previous reports and this work, demonstrating the modular and independent nature of the Rho-binding domain within Dia, which we have added to the manuscript:

•“Previous studies of the mDia1-Rho1 interaction showed that the BD is dispensable for the formation of an mDia1-Rho1 complex (31; 37).”

•We added an experiment that supports the notion of two distinct functional modules for Dia binding. Co-expression of GFP-Dia-N’ΔBD with *Rho1*-RNAi or expression of a GFP-Dia-N’ΔBD mut^G^ construct, resulted in complete loss of apical localization (Figure 7; quantification was added in 7G). This indicates that the residual apical localization observed for the GFP-Dia-N’ΔBD construct (Figure 6) can be attributed to Rho1 binding. A textual description was added to the Results section.

*A) The manuscript would be improved by the addition of immunoblotting to test the relative levels of FL versus ΔDAD*.

We added an immuno-blotting experiment showing similar levels of GFP-Dia-FL and GFP-DiaΔDAD expressed in embryos using *drm*-Gal4 in Figure 1—figure supplement 1. Live imaging of whole embryos expressing these constructs was added as well.

*B) The claim that the PIP2-Dia interaction is mediated by direct binding through the BD domain would be better substantiated by lipid flotation assays showing that the BD domain binds PIP2 and that the introduced point mutations abrogate binding. As these experiments take time and may not be possible in a reasonable time frame, the authors should at least acknowledge that direct binding was not tested*.

Preforming the suggested experiments was not possible in a reasonable time frame. However, we changed the wording to say: “*Drosophila* Dia also contains an N-terminal basic domain which is critical for apical localization (Figure 6), and *is likely to* mediate the association with PI(4,5)P2 through electrostatic interactions with the acidic phospholipid.”

*C) The results with exogenous PIP2 and MDCK cells strongly suggest that PIP2 alone is sufficient to direct Dia localization, if one assumes that there was no effect on RhoA localization in this experiment. Ideally RhoA localization should be examined in the MDCK assay; however, at the very least the authors should reconcile this result with their model, which suggests that even in the open conformation Dia requires interaction with Rho to localize properly*.

The MDCK system was used in this work to examine a causal role for PI(4,5)P2 in Dia recruitment. It was chosen because of the reported enrichment of PI(4,5)P2 in the apical domain and its critical role for lumen formation. We agree that in light of our model that was formulated for the *Drosophila* tissues, the dramatic apical localization of the GFP-DiaΔDAD construct in MDCK cysts could stem either from interactions only with PI(4,5)P2 (which may be exceptionally enriched in this system), or from an additional interaction with the heterologous RhoA. We do not know if such an interaction is possible, and if there is an apical bias for activated RhoA. A paragraph discussing this matter has been added to the Discussion.